# The Impact of Internal Marketing Practices on Employees' Job Satisfaction during the COVID-19 Pandemic: The Case of the Saudi Arabian Banking Sector

**Faisal Mohammed O. Almaslukh [1,\*], Haliyana Khalid [1,2] and Alaa Mahdi Sahi [1,3]**

[1] Azman Hashim International Business School, Universiti Teknologi Malaysia, Kuala Lumpur 54100, Malaysia; haliyana@utm.my (H.K.); sahi.alaa@graduate.utm.my (A.M.S.)

[2] College of Business Administration, University of Business and Technology, Jeddah 23435, Saudi Arabia

[3] College of Administration and Economics, Wasit University, Al-Kut 52001, Iraq

[\*] Correspondence: mofaisal@graduate.utm.my

**Abstract:** Based on the social exchange theory, the current study aimed to develop and test a conceptual model that integrates the relationships among internal marketing dimensions (i.e., supportive and participative leadership, training and development, information and communication, and selection and appointment) and job satisfaction in the banking sector of Saudi Arabia, particularly during the COVID-19 pandemic. We collected data from 329 employees working in different private and public banks in Riyadh, Saudi Arabia. Overall, the findings confirmed the significant and positive effects of supportive and participative leadership, training and development, information and communication, and selection and appointment on employees' job satisfaction. The current research contributes to the understanding of the broad role played by internal marketing practices in maintaining the job satisfaction of banking sector employees, during and possibly after the COVID-19 pandemic.

**Keywords:** internal marketing; job satisfaction; banking sector; social exchange theory; Saudi Arabia

## 1. Introduction

The COVID-19 pandemic has pushed all economic sectors into a complex and unpredictable situation. Currently, it is challenging for most sectors to identify or create new business opportunities in domestic and international markets [1]. Despite being an essential pillar of a country's economic development [2,3], the banking sector has especially faced difficulties in enhancing and maintaining its market presence, productivity, and performance following the pandemic [4–7]. Notably, the intensifying competition among banks makes it not only more complex but also more imperative for banks to create value for external and internal stakeholders in order to survive, both during and after the pandemic [8,9]. In particular, the employees of an organization are crucial in enhancing the organization's performance in the present climate.

Employees are among a company's pre-eminent assets and serve as productive liaisons with external audiences, i.e., customers [10]. Prior studies have found that, in service organizations such as banks, employees are valuable stakeholders whose engagement and satisfaction companies strive to increase by developing broader strategies [11–15]. Empirically, scholars have concluded that employees' positive behaviors and attitudes enhance their satisfaction; as a result, higher customer engagement is achieved [16–18]. Several researchers have further confirmed that employees' job satisfaction is broadly influenced by their service quality [19–22]. In other words, frontline employees of banks are agents who deliver high-quality services and create a supportive relationship with customers [23]. This connection is a key determinant of high customer satisfaction and loyalty [24].

Retaining and motivating employees is therefore crucial for boosting the productivity and performance of the banking sector. Specifically, it is important for the banking sector

to identify and understand the key factors that make employees satisfied and motivated to perform their work responsibilities, as Mohammad et al. pointed out that satisfied employees are more productive [25]. In fact, Putra et al. noted that several banks have begun to pay attention to the indicators of employees' satisfaction and engagement with their job [10].

To this end, we considered the importance and applicability of internal marketing practices as a tool to engender employees' job satisfaction during and after the COVID-19 pandemic, which could potentially enable banks to compete in the post-pandemic market [26]. Internal marketing refers to "viewing employees as internal customers, viewing jobs as internal products that satisfy the needs and wants of these customers, while addressing the objectives of the firms" [27]. In essence, internal marketing practices hand over a firm to its employees [28]. However, there is a lack of research on internal marketing practices in developing countries [29], with most extant research having been conducted in developed countries such as the USA and the UK. Moreover, the concept of internal marketing is relatively new in the banking sector of developing countries. Oladapo et al. reported that banking institutions in such countries, including Saudi Arabia, still practice traditional systems to train employees to deal with customers [30].

More recently, Alzaydi concluded that the banks in Saudi Arabia failed to sustain the quality of services during the COVID-19 pandemic, as before the pandemic, they were offering effective services to customers [31]. Furthermore, the consumers' attitude and behavior during the COVID-19 pandemic changed, and they started demanding a more modern service quality [32–34].

Baz et al. similarly highlighted that banks are yet to implement a modern customer management system due to a lack of technological innovation [35]. Consequently, it is essential for banking organizations to understand their context-specific customer culture and, subsequently, to educate their employees accordingly via effective internal marketing practices so that they can effectively handle and satisfy customers [5,10,27,36]. Therefore, several empirical studies investigated and concluded that the banks in Saudi Arabia are still practicing the traditional customer service style [30]. Thus, Akbari highlighted that the leadership of the banking sector is not designing and practicing modern customer services practices, mainly in developing countries [37]. In this regard, Tuuk suggested that the leadership of the banks should discuss with and allow the frontline employees to participate in designing the customer services strategies, as these employees can understand and predict the customers' intentions and attitudes to be satisfied [38].

To bridge the aforementioned gap in the literature, we aimed to investigate the impact of four internal marketing dimensions (i.e., supportive and participative leadership, training and development, information and communication, and selection and appointment) on Saudi Arabian bank employees' job satisfaction during the COVID-19 pandemic. We applied the social exchange theory (SET) [39] to achieve the study objectives. According to Sohail and Jang, the SET "implies that a satisfied employee extends his level of satisfaction to a customer as a way of showing appreciation or gratitude to his employer, who, in one way or another, provides satisfactory working conditions to the employee" [22].

This study contributes to the literature on internal marketing practices in several ways. First, it identifies and explains the role of the critical indicators of internal marketing (i.e., supportive and participative leadership, training and development, information and communication, and selection and appointment) by verifying their impact on employees' job satisfaction in the banking sector. Previous studies have largely focused on the direct impacts of internal marketing practices on the behaviors and attitudes of employees in different organizations [22,40–42], neglecting the underlying mechanisms through which these effects take place. The third contribution of this study is the application of the SET to develop the research framework, which was subsequently tested using a structural equation modeling (SEM) analysis.

This paper is organized as follows: first, we present a review of relevant empirical studies before identifying the dimensions of internal marketing practices and developing

the theoretical ground of the study. Following that, we discuss our research methods, samples, and measures. The results of the analysis are then reported and interpreted. Finally, a discussion of the study's implications, limitations, and future research directions conclude the paper.

*Country Context—Saudi Arabia*

According to a recent report published by the Saudi Arabian Monetary Authority (SAMA), a total of 25 commercial banks, including 13 domestic and 12 international banks, are operating in different cities of Saudi Arabia [43]. The banking sector in Saudi Arabia is growing rapidly and appreciably, contributing to the development of the economy and human capital [30,32,44]. By total assets, Saudi Arabia's banking sector ranks second in the Gulf Cooperation Council (GCC) countries and comprises over 29% of all banking assets in the region; this is despite making up around 34% of the banking sector's total credit, lending to commerce, manufacturing, and construction sectors on the country [45].

## 2. Literature Review and Theoretical Development

Internal marketing is a philosophy that advocates a modern view of employees. According to Chen and Sriphon, the leadership of any organization typically plays an important role in identifying and implementing the latest trends that attain business goals and sustain competitive advantages [46]. In this regard, the internal marketing approach involves organizational leadership, treating employees as customers [37] by prioritizing their satisfaction and engagement [47,48]. Nemteanu and Dabija suggested that organizations should practice internal marketing as it can significantly engage and satisfy employees in terms of their job environment [49]. This is especially pertinent during the COVID-19 pandemic, when the banking sector is challenged to develop a safe yet socially distanced connection between employees and customers to minimize the spread of the virus [5,50]. For example, in Saudi Arabia, the Saudi Arabian Monetary Authority (SAMA) introduced COVID-19 SOPs for the banking sector and instructed all financial institutions to maintain social distancing with their customers [51,52]. In such a challenging crisis, banking organizations need to keep their employees motivated and satisfied so that they can work effectively and contribute to achieving business goals. Accordingly, internal marketing can be considered a critical practice that sustains employees' job satisfaction [27,53]. Specifically, it has been confirmed by several authors that internal marketing practices include four key dimensions (i.e., supportive and participative leadership styles, training and development, information and communication, and selection and appointment) that can significantly predict employees' job satisfaction [22].

According to Sohail and Jang "internal marketing leads to employee satisfaction; employee satisfaction leads to service quality; service quality leads to customer satisfaction; customer satisfaction leads to repeat purchases of the service, and repeated purchases lead to organizational profitability" [22]. Indeed, COVID-19 has given rise to various employee implications that can be explored with the SET, including exhaustion, burnout, anxiety, stress, performance, counterproductive work behaviors, health, development, well-being, and satisfaction [49]. The SET posits that organizations that offer their employees a satisfactory working environment amid the pandemic can derive a high level of service quality [20,21] because employees' gratitude and appreciation to the organization will be reflected in their services to customers [28,54]. Furthermore, scholars have applied the SET to establish the impact of internal marketing practices on employees' job satisfaction [22,49,55]. Despite being touted as one of the best theories to support employees' job satisfaction, few scholars have extended the SET to investigate the impact of internal marketing practices on employees' job satisfaction in the banking sector. To bridge this theoretical gap, we developed a research model integrating four internal marketing dimensions (i.e., supportive and participative leadership styles, training and development, information and communication, and selection and appointment), and job satisfaction. The following sections present the development of the hypotheses and research model (see Figure 1).

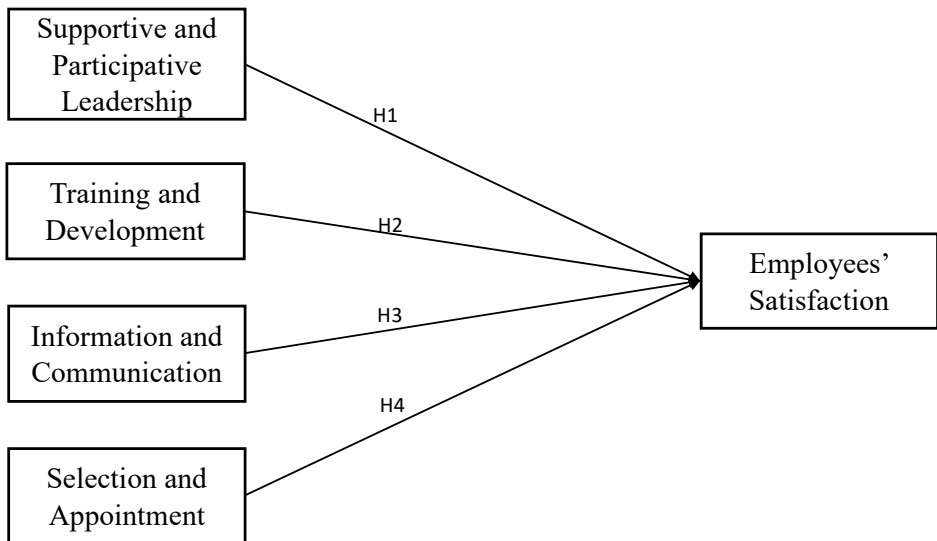

**Figure 1.** Research framework.

### 2.1. Supportive and Participative Leadership and Employees' Job Satisfaction

According to Vrontis et al. supportive and participative leadership, as a key dimension of internal marketing in an organization, is liable to design and implement modern strategies and market culture trends that motivate employees to work [56]. Managers of an organization can implement supportive and participative leadership styles to enhance employees' job satisfaction [22,57,58]. Specifically, senior managers can educate and monitor frontline employees who directly interact with customers [37], as these managers are generally role models for their immediate subordinates. By doing so, managers can motivate and empower their subordinates to work on new business ideas, as well as to deal with customers effectively.

Prior studies have confirmed that supportive and participative leadership significantly influences employees' job satisfaction [59,60]. More recently, Sohail and Jang found that supportive leadership in an organization plays an important role in educating employees on how to interact with customers, thereby enhancing the quality of the services [22]. Likewise, Tuuk confirmed that effective supportive leadership in the banking sector significantly improves employees' job satisfaction, which is likely during the COVID-19 pandemic as well [38]. Considering the above discussion, we proposed the following hypothesis:

**H1:** *Supportive and participative leadership significantly and positively impacts employees' job satisfaction.*

### 2.2. Training and Development and Employees' Job Satisfaction

Nemteanu and Dabija pointed out that the COVID-19 pandemic has created serious and unpredictable challenges for organizations [49]. Amid this crisis, organizations face high market competition [5,61], such that their success or failure depends on the ability of skilled employees to manage and resolve growing challenges [13]. Skilled employees are assumed to provide dynamic services and contribute to the organization's productivity, mainly by interacting with customers effectively [25]. Therefore, Khanfar concluded that employee training and development programs significantly enhance organizations' overall performance and quality of services [62]. Accordingly, Musa et al. suggested that organizations should arrange updated training and development programs for their employees, since the COVID-19 pandemic has changed the way of doing business [63].

Empirical findings have confirmed that training and development is a dominant dimension of internal marketing [53]. Adding or extending knowledge is one of the important factors that impels organizations to attain business goals and sustain competitive advantages [62,64]. Recently, Shen and Tang, and Jaworski et al. found a strong positive

relationship between the training and development of employees and their job satisfaction [65,66]. Similarly, Arroyo-López et al. corroborated that employees' training and development programs enable them to work effectively and maintain a high quality of services [67]. Thus, we developed the following hypothesis:

**H2:** *Training and development significantly and positively impacts employees' job satisfaction.*

### 2.3. Information and Communication and Employees' Job Satisfaction

According to Zoe and Ogba, and Sarker and Ashrafi, information and communication is among the most common dimensions of internal marketing in service organizations [68,69]. Considering the COVID-19 crisis, the frontline staff of the banking sector apply this dimension to interact with external stakeholders, i.e., customers. On the other hand, organizational leadership also practices internal information and communication to reach the organization's internal customers, i.e., employees [22]. Putra et al. observed that, during the COVID-19 pandemic, it was a challenging task for employees to communicate with customers while maintaining the quality of services amid enforced SOPs [10]. Management must therefore understand and implement modern communication applications for employees to effectively interact with customers; as a result, quality of services can be maintained for the long term [37,60,64,70], and employees can also feel satisfied with their job [22].

Further, Rumiyati and Syafarudin, described the severe post-COVID-19 impacts on the banking sector due to high market competition [71]. In this regard, Marcu suggested that the banking sector should effectively apply internal marketing practices to sustain the performance of employees and competitive advantages in the market [5,21]. A modern communication system that integrates external and internal communication and enables employees to interact with and serve customers effectively could therefore lead to an efficient quality of services [68,69,72–74]. This is because a practical and modern communication system boosts employees' confidence and assures them of their service delivery. In line with this discussion, we proposed the following hypothesis:

**H3:** *Information and communication significantly and positively impacts employees' job satisfaction.*

### 2.4. Selection and Appointment and Employees' Job Satisfaction

According to Mehrabad and Brojeny, and Easterby-Smith et al. the term 'selection and appointment' was introduced by human resource management in the 1980s to represent a significant change in an organization's strategic direction [75,76]. Researchers have identified selection and appointment (staffing) as a common dimension of internal marketing [53,59]. Internal marketing practices guide organizational leadership to hire skilled, reliable, qualified, and capable employees who would feel more satisfied in the working environment, leading to a dynamic quality of the services [77,78]. It is even more critical to hire skilled employees under the COVID-19 pandemic as they can deliver and maintain maximum services to customers. In fact, COVID-19 has changed the trends of performing a particular job. However, researchers have paid less attention to the dimension of selection in the banking sector. Empirically, limited studies have been conducted in developing countries to investigate the impact of selection and appointment in an organization, especially in the banking sector. Thus, we developed the following hypothesis:

**H4:** *Selection and appointment significantly and positively impacts employees' job satisfaction.*

### 3. Research Framework

We developed a research framework for the current study, pointing to the empirical evidence on internal marketing practices and employees' job satisfaction. Social exchange theory was applied to understand internal marketing dimensions, i.e., (supportive and participative leadership, training and development, information and communication, and selection and appointment) and employees' job satisfaction in the banking sector of Saudi Arabia. Thereby, in this framework, supportive and participative leadership, training and

development, information and communication, and selection and appointment exert a direct link with employees' job satisfaction presented in H1, H2, H3, and H4. Thus, the research framework is presented in Figure 1.

## 4. Research Methodology

### 4.1. Research Design and Context

The current study aimed to investigate the impact of internal marketing dimensions (i.e., supportive and participative leadership styles, training and development, information and communication, and selection and appointment) on employees' job satisfaction during the COVID-19 pandemic. The quantitative approach was adopted to test the research model in the banking sector of Saudi Arabia. This research context was justified because employees' job satisfaction in the banking sector has been negatively affected by the COVID-19 pandemic. According to Putra et al. employees in the banking sector are neither satisfied nor motivated to work under the restrictions of the pandemic. Employees now have to spend the majority of their time at the banks strictly following SOPs to avoid spreading the virus [10].

We developed a self-administered questionnaire to collect data from the targeted respondents, i.e., Saudi Arabian bank employees. Before the actual data collection, we carried out a pilot test. According to Urbach and Ahlemann, this pilot test process supports the researchers in evaluating the measurement instrument in terms of format, ease of understanding, content, etc. [79]. Thus, in the present study, a pilot study was conducted with a sample of 20 employees working in the banking sector of Saudi Arabia.

According to a recent report by SAMA, a total of 25 public and private commercial banks are operating in Saudi Arabia [80], most of which have several branches in the capital city of Riyadh. We thus approached 20 banks in Riyadh for data collection, but only received approval to collect data from 13 banks (see Table 1). From these banks, we collected data from 340 respondents using cross-sectional random sampling [81]. After excluding 11 questionnaires for straight-lining answers and/or missing data, the final sample size for data analysis was 329.

**Table 1.** Demographic information.

|  | **Frequency** | **Percent** |
|---|---|---|
| Gender | | |
| Male | 211 | 64.1 |
| Female | 118 | 35.9 |
| Age (Years) | | |
| 18–25 | 22 | 6.7 |
| 26–35 | 93 | 28.3 |
| 36–45 | 126 | 38.3 |
| 46–55 | 66 | 20.1 |
| More than 55 years | 22 | 6.7 |
| Education | | |
| Bachelor's | 112 | 34 |
| Master's | 173 | 52.6 |
| Doctorate | 4 | 1.2 |
| Other | 40 | 12.2 |
| Your Current Position in The Bank | | |
| Operation manager | 35 | 10.6 |
| Credit or loan officer | 70 | 21.3 |
| Customer relationship manager | 84 | 25.5 |
| Frontline officer | 125 | 38 |
| Insurance manager | 12 | 3.6 |
| Remittance manager | 3 | 0.9 |

**Table 1.** *Cont.*

|  | Frequency | Percent |
|---|---|---|
| **Your Current Bank of Employment** | | |
| The National Commercial Bank | 83 | 25.2 |
| The Saudi British Bank | 12 | 3.6 |
| Saudi Investment Bank | 8 | 2.4 |
| Alinma Bank | 26 | 7.9 |
| Banque Saudi Fransi | 12 | 3.6 |
| Riyad Bank | 38 | 11.6 |
| Samba Financial Group (Samba) | 18 | 5.5 |
| Alawwal Bank | 23 | 7 |
| Al Rajhi Bank | 17 | 5.2 |
| Arab National Bank | 29 | 8.8 |
| Bank AlBilad | 13 | 4 |
| Bank AlJazira | 28 | 8.5 |
| Gulf International Bank Saudi Arabia (GIB-SA) | 22 | 6.7 |
| **Your Department** | | |
| Operations | 93 | 28.3 |
| Credit and loan | 140 | 42.6 |
| Customer relationship | 35 | 10.6 |
| Front officers (account opening) | 32 | 9.7 |
| Insurance | 15 | 4.5 |
| Remittance | 14 | 4.3 |
| **Experience in the Current Position** | | |
| Less than one year | 19 | 5.8 |
| 1–5 years | 64 | 19.5 |
| 6–10 years | 110 | 33.4 |
| 11–15 years | 115 | 35 |
| More than 15 years | 21 | 6.4 |

The respondents were employees working in different public and private banks in Riyadh, Saudi Arabia. From the 329 respondents, 64.1% were male and 35.9% were female. The dominant age group was 36 to 45 year old's (38.3%), while more than half (52.6%) of the respondents held a master's degree. The majority of respondents were frontline officers (38%), employed at The National Commercial Bank (25.2%), and from the credit and loan department (42.6%). In terms of work experience in their current job position, most respondents had worked for six to 10 years (33.4%) in the same department. The detailed demographic profile of the respondents is presented in Table 1. And descriptive statistics values of the constructs are presented in Table 2.

**Table 2.** Descriptive Statistics.

| Constructs | Mean | SD |
|---|---|---|
| Supportive and Participative Leadership | 4.3444 | 0.37490 |
| Training and Development | 4.6052 | 0.45382 |
| Information and Communication | 4.5350 | 0.34571 |
| Selection and Appointment | 4.4101 | 0.32798 |
| Job Satisfaction | 5.5043 | 0.43832 |

*4.2. Measures*

The research model encompassed six variables that were assessed in the questionnaire. The measurement items for all of the variables were adapted from previous studies [22,69] and modified as per the current study's context (see Table 3). Items for the dimensions of internal marketing (i.e., supportive and participative leadership styles, training and development, information and communication, and selection and appointment), and job

satisfaction were adapted from [22]. A five-point Likert scale (1 = strongly disagree to 5 = strongly agree) was utilized to solicit the respondents' actual opinions on the items.

**Table 3.** Reliability and convergent validity results.

| | Loading | Alpha | CR | AVE |
|---|---|---|---|---|
| Supportive and Participative Leadership | | 0.737 | 0.775 | 0.565 |
| The bank management supports me in achieving the company's vision | 0.643 | | | |
| The bank management supports me in performing well | 0.699 | | | |
| My senior management consults me if I face problems | 0.623 | | | |
| My senior management consult me if I face problems | 0.754 | | | |
| Training and Development | | 0.765 | 0.810 | 0.592 |
| The bank management supports me in the development of knowledge and skills | 0.602 | | | |
| My skills and knowledge development are an ongoing process in this bank | 0.827 | | | |
| The bank goes beyond training and educates me | 0.855 | | | |
| Information and Communication | | 0.714 | 0.821 | 0.535 |
| The bank places considerable emphasis on communication with me | 0.723 | | | |
| The bank communicates with me for the importance of my job roles | 0.727 | | | |
| The bank's internal communication is consistent with our advertising | 0.687 | | | |
| The bank gathers data from employees and uses it to improve job performance | 0.787 | | | |
| Selection and Appointment | | 0.769 | 0.775 | 0.536 |
| The bank takes great effort to select the right person | 0.755 | | | |
| The bank places considerable importance on the staffing process | 0.714 | | | |
| The bank's long-term employee potential is emphasized | 0.726 | | | |
| Job Satisfaction | | 0.709 | 0.820 | 0.533 |
| I am satisfied with my nature of job | 0.750 | | | |
| I am satisfied with my relationship with my fellow workers | 0.784 | | | |
| I am satisfied with the supervision of my superior | 0.713 | | | |
| I am satisfied with my salary in this bank | 0.667 | | | |

### 4.3. Data Analysis

We applied partial least squares structural equation modeling (PLS-SEM) via the SmartPLS software to analyze the data. According to Saleem et al., social and management science researchers widely apply SEM approaches to evaluate the reliability and validity of their research model [82]. PLS-SEM analysis was used in this study in two assessment stages (i.e., measurement model and structural model) to examine the direct and indirect relationships between the constructs, evaluate the path coefficients, and justify the theoretical propositions [81].

### 4.4. Measurement Model Results

Under the measurement model, the constructs' reliability and convergent validity were evaluated using Cronbach's alpha ($\geq$0.50), composite reliability (CR) ($\geq$0.60), and average variance extracted (AVE) ($\geq$0.50). As can be seen in Table 3, the loadings of all the items measured above the threshold value of 0.6; thus, items with lower values were deleted [70]. The values of Cronbach's alpha, CR, and AVE for all items also exceeded the required criteria, confirming the constructs' reliability and convergent validity. Tables 4 and 5 further show that the constructs fulfilled the HTMT and Fornell–Larcker criterion of discriminant validity.

**Table 4.** Discriminant validity (Fornell–Larcker criterion).

|   |   | 1 | 2 | 3 | 4 | 5 |
|---|---|---|---|---|---|---|
| 1 | Employees' Satisfaction | 0.730 | | | | |
| 2 | Information and Communication | 0.511 | 0.732 | | | |
| 3 | Selection and Appointment | 0.460 | 0.548 | 0.732 | | |
| 4 | Supportive and Participative Leadership | 0.506 | 0.64 | 0.575 | 0.682 | |
| 5 | Training and Development | 0.389 | 0.483 | 0.45 | 0.429 | 0.770 |

**Table 5.** Discriminant validity (HTMT).

|   |   | 1 | 2 | 3 | 4 |
|---|---|---|---|---|---|
| 1 | Information and Communication | 0.683 | | | |
| 2 | Selection and Appointment | 0.704 | 0.870 | | |
| 3 | Supportive and Participative Leadership | 0.681 | 0.770 | 0.856 | |
| 4 | Training and Development | 0.534 | 0.675 | 0.742 | 0.667 |

*4.5. Structural Model Results*

Once the measurement model established the constructs' validity and reliability, the bootstrapping procedure with 5000 subsamples was used to obtain the structural model (see Figure 2). Moreover, there are several other statistical criteria to validate the structure model including coefficient of determination ($R^2$), cross-validated redundancy ($Q^2$), path coefficients, and the effect size ($f^2$) [83,84].

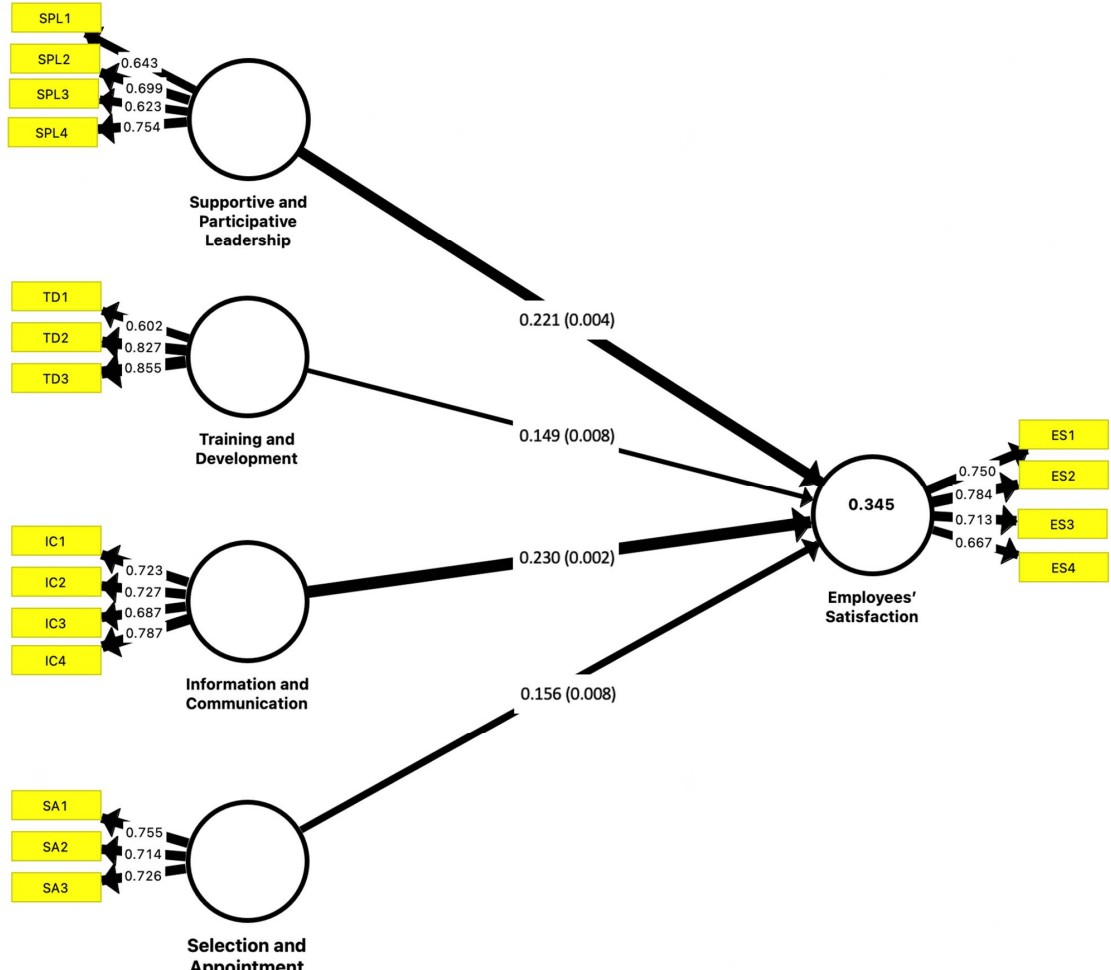

**Figure 2.** Structural model.

As per the direct relationship results shown in Table 6 supportive and participative leadership, training and development, selection and appointment, and information and communication were found to have significant positive effects on employees' job satisfaction. Therefore, H1, H2, H3, and H4 were supported. The $R^2$ value for employees' job satisfaction (0.345) was relatively high, suggesting that the internal marketing dimensions explained a large amount of the variance in employees' job satisfaction. Furthermore, the $Q^2$ value of employees' job satisfaction was 0.171, which suggested a medium predictive relevance. Thus, the values are presented in Table 7.

**Table 6.** Direct relationship results.

|  |  | Original Sample | T Statistics | *p* Values |
|---|---|---|---|---|
| H1 | Supportive and Participative Leadership → Employees' Job Satisfaction | 0.221 | 2.860 | 0.004 |
| H2 | Training and Development → Employees' Job Satisfaction | 0.149 | 1.885 | 0.008 |
| H3 | Information and Communication → Employees' Job Satisfaction | 0.230 | 3.182 | 0.002 |
| H4 | Selection and Appointment → Employees' Job Satisfaction | 0.156 | 2.650 | 0.008 |

**Table 7.** Predictive validity of constructs ($Q^2$).

| Construct | Redundancy |
|---|---|
| Employees' Job Satisfaction | 0.171 |

Furthermore, in this study, Table 7 shows the $f^2$ values for each path. For example, path supportive and participative leadership → employees' job satisfaction, training and development → employees' job satisfaction, information and communication → employees' job satisfaction, and selection and appointment → employees' job satisfaction showed the medium effect size. Thereby, the detailed results are presented in Table 8.

**Table 8.** $f^2$ Values for each Path.

| Path | Effect Size |
|---|---|
| Supportive and Participative Leadership → Employees' Job Satisfaction | 0.041 |
| Training and Development → Employees' Job Satisfaction | 0.022 |
| Information and Communication → Employees' Job Satisfaction | 0.038 |
| Selection and Appointment → Employees' Job Satisfaction | 0.014 |

## 5. Discussion and Conclusions

We developed a theoretical model to investigate the direct relationships between internal marketing practices (i.e., supportive and participative leadership, training and development, information and communication, and selection and appointment), and employees' job satisfaction. Our findings reveal that a supportive leadership style positively impacts employees' job satisfaction (β = 0.221, *t*-value = 2.860, *p*-value = 0.004), which is consistent with previous studies, such as that of [85]. Amid the COVID-19 pandemic, the banking sector faces numerous market challenges [26]; at this juncture, supportive leadership can play an important role in introducing or adapting a new business trend, and creating a normal working environment for employees to feel more motivated and provide the maximum quality of services [71]. Recently, Haque pointed out that, during the COVID-19 pandemic, the organizational leadership played a robust role in learning and implementing modern business trends, as it can maintain and deliver effective service quality to the consumers [86].

The results also showed a robust positive relationship between training and development and employees' job satisfaction (β = 0.149, *t*-value = 1.885, *p*-value = 0.008). Several

studies have concluded that training and development is a crucial dimension of the internal marketing because it is a unique resource that supports organizational leadership, not only in developing modern business strategies, but also in training new and old employees to achieve business goals and make them satisfied to work more effectively. Since the COVID-19 pandemic, it has been challenging for banking sector employees to learn and implement new norms of operations and provide essential services to customers while maintaining quality [49]. In this case, training and development can support employees and teach them new trends of customer management, which would sustain the performance from all aspects [65]. Accordingly, Prasetyaningtyas et al. highlighted that, during the COVID-19 pandemic, companies worldwide, including the banking sector, introduced a new working style called work from home (WFH) and forced their employees to work regularly from home [87]. Thus, due to the novelty of the WFH concept, most employees were unaware of how to effectively WFH [87]. In this regard, Zito et al. suggested that organizational leadership should offer modern training and development programs to their employees so that they can be more satisfied with the job while WFH [88].

We also found a positive and strong relationship between information and communication and employees' job satisfaction ($\beta$ = 0.230, *t*-value = 3.182, *p*-value = 0.002). Scholars have cited information and communication as the best predictor of employees' job satisfaction [89]. Sanders et al. pointed out that, during the COVID-19 pandemic, information and communication between employees and customers was an arduous task to perform [90]. Further, Sohail and Jang stated that when "employees with multicultural orientations and/or different ethnic backgrounds, different nationalities and different demographic characteristics come together, a tendency towards miscommunication in both verbal and nonverbal communication is likely to occur" [22]. Nonetheless, it appears that better communication systems during and after the pandemic are imperative for quality service delivery.

The positive relationship between selection and appointment and employees' job satisfaction was supported as well ($\beta$ = 0.156, *t*-value = 2.650, *p*-value = 0.008), corroborating the work of Khan et al. and Barkhuizen et al. Further, they reported that during the COVID- 19 pandemic, organizational leadership needs to hire qualified, skilled, and experienced candidates to be the "right person for the right job," as these candidates can fill the organizational communication gaps caused by the pandemic [91,92].

The findings of this study add to the literature by identifying the specific internal marketing dimensions (i.e., supportive and participative leadership, training and development, information and communication, and selection and appointment) that enhance bank employees' job satisfaction in the COVID-19 context. This study contributes new knowledge that bridges internal marketing research and job satisfaction research. Finally, we have extended the application of the SET to the context of internal marketing in the banking sector of a developing country, particularly during a global pandemic.

## 6. Limitations and Future Study

Our research contains certain limitations and recommendations for future research. First, we applied the cross-sectional method to obtain data from the respondents during the pandemic. Future researchers are suggested to conduct a longitudinal study using the same research model and theory to compare the long-term effects of the pandemic on employee behaviors. Next, the current study was limited to banks in Riyadh, Saudi Arabia. Future studies are advised to carry out the same research in different countries, cultures, and economic sectors. This will add new knowledge to the literature on internal marketing practices and employees' job satisfaction under the lens of the COVID-19 pandemic.

**Author Contributions:** Conceptualization, F.M.O.A. and H.K.; Data curation, F.M.O.A. and A.M.S.; Formal analysis, F.M.O.A., H.K. and A.M.S.; Funding acquisition, F.M.O.A.; Investigation, F.M.O.A. and H.K.; Methodology, F.M.O.A. and H.K.; Project administration, H.K.; Resources, A.M.S.; Software, F.M.O.A. and A.M.S.; Supervision, H.K.; Validation, F.M.O.A., H.K. and A.M.S.; Visualization, H.K.; Writing—original draft, F.M.O.A.; Writing—review & editing, F.M.O.A. All authors have read and agreed to the published version of the manuscript.

**Funding:** This research received no external funding.

**Institutional Review Board Statement:** Not applicable.

**Informed Consent Statement:** Not applicable.

**Conflicts of Interest:** The authors declare no conflict of interest.

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
