# Peer review of "The Impact of Internal Marketing Practices on Employees’ Job Satisfaction during the COVID-19 Pandemic: The Case of the Saudi Arabian Banking Sector"

_sustainability, doi:10.3390/su14159301_

Round 1

Reviewer 1 Report

The article is interesting and well constructed, but needs some additional work before it is publishable.

1. The theoretical framework is good, the methodology adequate, although the question of the novelty of the study could be questioned since all the scales come from the same author, which shows little theoretical contribution. Clearly a point to be considered in the limitations. Another limitation is the measurement of service quality from employees' perceptions, which is a measurement subject to a strong bias.

2. The authors are not developing the necessary tests for the use of PLS, and the analysis of indicators such as HTMT and Q2 is missing.

3. There is a central issue that should be considered by the authors. If it is not resolved, the model does not have the quality necessary to be published. I am referring to the coefficient of determination R2 of the dependent variable which is less than 0.1, being a disappointing value. The authors should try to solve this problem, for example, by considering in the model the direct relations between the independent variables and the dependent one and see if this improves that coefficient. This implies a revision of the hypotheses and theoretical framework.

Reviewer 2 Report

The article presents a high level of academic writing. The structure is consistent and the research framework is clear. 

I  recommend only minor revision concerns with: 

1. In the introduction more attention to explaining how a pandemic situation affected service quality and employee satisfaction, especially in the local context should be put.

2. It will also be useful to shortly characterize the banking sector of Saudi Arabia and indicate if it is different from other countries.

3. In lines 326-327 authors indicate that "all variables were adapted from previous studies" but they do not specify in what studies. References are needed.

4. In present form figure, 2 is unreadable.

5. Table with the main statistics (M, SD and correlations)  of the analysed variable will be useful.

6. In the discussion, analysing how pandemic situations affect the results should be paid more attention. Did analysed relations different because of COVID?

Reviewer 3 Report

The paper is extremely well-written and although it presents topics that have been well-covered in the literature, the research gap, and basis for needing this study, is well-defended through the context of the study. 

The methodology is clear, with a breakdown of the sample and testing their measures for reliability. Results are clear and discussion covered the key issues, alongside relevance recommendations and directions for further research.  

Round 2

Reviewer 1 Report

This issue is not properly solved: "3. There is a central issue that should be considered by the authors. If it is not resolved, the model does not have the quality necessary to be published. I am referring to the coefficient of determination R2 of the dependent variable which is less than 0.1, being a disappointing value. The authors should try to solve this problem, for example, by considering in the model the direct relations between the independent variables and the dependent one and see if this improves that coefficient. This implies a revision of the hypotheses and theoretical framework."

This is not a proper answer: " We took a close look of coefficient of determination R2, all the values extracted from the data we collected. Furthermore, we provided descriptive statistics (table 2) to justify our findings. "

Please be serious about this topic. If not properly solved I'll recommend article rejection

Round 3

Reviewer 1 Report

Thank you for the revision